# A scalable and tunable platform for functional interrogation of peptide hormones in fish

Eitan Moses[1], Roman Franek[1,2], Itamar Harel[1]*

[1]Department of Genetics, the Silberman Institute, The Hebrew University of Jerusalem, Jerusalem, Israel; [2]University of South Bohemia in Ceske Budejovice, South Bohemian Research Center of Aquaculture and Biodiversity of Hydrocenoses, Vodnany, Czech Republic

**Abstract** Pituitary hormones play a central role in shaping vertebrate life history events, including growth, reproduction, metabolism, and aging. The regulation of these traits often requires precise control of hormone levels across diverse timescales. However, fine tuning circulating hormones in-vivo has traditionally been experimentally challenging. Here, using the naturally short-lived turquoise killifish (*N. furzeri*), we describe a high-throughput platform that combines loss- and gain-of-function of peptide hormones. Mutation of three primary pituitary hormones, growth hormone (*gh1*), follicle stimulating hormone (*fshb*), and thyroid stimulating hormone (*tshb*), alters somatic growth and reproduction. Thus, suggesting that while the killifish undergoes extremely rapid growth and maturity, it still relies on vertebrate-conserved genetic networks. As the next stage, we developed a gain-of-function vector system in which a hormone is tagged using a self-cleavable fluorescent reporter, and ectopically expressed in-vivo through intramuscular electroporation. Following a single electroporation, phenotypes, such as reproduction, are stably rescued for several months. Notably, we demonstrate the versatility of this approach by using multiplexing, dose-dependent, and doxycycline-inducible systems to achieve tunable and reversible expression. In summary, this method is relatively high-throughput, and facilitates large-scale interrogation of life-history strategies in fish. Ultimately, this approach could be adapted for modifying aquaculture species and exploring pro-longevity interventions.

*For correspondence:
itamarh@mail.huji.ac.il

Competing interest: The authors declare that no competing interests exist.

## Editor's evaluation

Moses and Harel generate a compelling set of novel molecular tools in African turquoise killifish, including the development of gain of function and dose dependent inducible expression systems for African turquoise killifish. These tools will help boost this budding model system for broad biotechnological applications, including the study of gene function in the context of aging in a relatively fast manner compared to other vertebrate models. The authors showcase the efficacy of their tools in the context of peptide hormones involved in growth and gonad development.

## Introduction

Pituitary hormones are key regulators of vertebrate life-history events, including growth, reproduction, metabolism, and aging (*Figure 1A*). In many cases, regulation of these traits requires precise control of hormone levels over a diverse range of temporal scales (*Taylor et al., 2019*). While genetic perturbations of pituitary hormones have been instrumental in understanding their function (*Liu and Lin, 2017*; *Hu et al., 2019*), experimental approaches primarily depend on whole-body loss-of-function.

**eLife digest** In humans and other vertebrates, a pea-size gland at the base of the brain called the pituitary gland, produces many hormones that regulate how individuals grow, reproduce, and age. Three of the most prominent hormones are known as the growth hormone, the follicle-stimulating hormone, and the thyroid-stimulating hormone.

It is important that the body precisely controls the levels of these hormones throughout an individual's life. One way researchers can investigate how hormones and other molecules work is to artificially alter the levels of the molecules in living animals. However, this has proved to be technically challenging and time-consuming for pituitary gland hormones.

Moses et al. studied the growth hormone, follicle-stimulating hormone, and thyroid-stimulating hormone in the turquoise killifish, a small fish that grows and matures more rapidly than any other vertebrate research model. The experiments revealed that mutant fish lacking one of the three primary pituitary hormones were smaller, took longer to reach maturity, or were completely sterile. This suggests these three hormones play a similar role in killifish as they do in other vertebrates.

The team then developed a new experimental platform to precisely control the levels of the three hormones in killifish. Genes encoding individual hormones were expressed in the muscles of the mutant fish, effectively making the muscles a 'factory' for producing that hormone. Treating mutant fish this way once was enough to restore growth and to fully return reproduction to normal levels for several months. Moses et al. also demonstrated that it is possible to use this platform to express more than one hormone gene at a time and to use drugs to switch hormone production on and off in a reversible manner. For example, this reversible approach made it possible to effectively adjust fertility levels.

The new platform developed in this work could be adapted for modifying a variety of traits in animals to explore how they impact health and longevity. In the future, it may also have other applications, such as optimizing how farmed fish grow and reproduce and regulating hormone levels in human patients with hormone imbalances.

---

As a result, interpretations are subject to bias due to stage-dependent hormonal requirements, or potential compensatory mechanisms. Thus, developing reversible control of pituitary hormones is imperative for a mechanistic understanding. Gain-of-function methodologies, primarily in fish, have so far depended on laborious and repeated injections of pituitary extracts (which are non-specific), or require purification of recombinant hormones (which is time consuming; *Mylonas et al., 2010*; *Taranger et al., 2010*). Such challenges significantly hamper the interpretation and scalability of these experiments.

Zebrafish (*Danio rerio*) and medaka (*Oryzias latipes*) are the most widely used genetic fish models. Interestingly, while these versatile models share a range of experimental advantages, they also exhibit several characteristics that are less compatible with high-throughput exploration of adult physiology. For example, both fish undergo relatively slow sexual maturation (~3–4 months), and depend on the germline to develop into phenotypic females (*Slanchev et al., 2005*; *Kurokawa et al., 2007*). Accordingly, perturbations that affect the germline, such as mutations in the follicle stimulating hormone receptor gene (*fshr*), produce an all-male sex-reversal (*Slanchev et al., 2005*; *Kurokawa et al., 2007*) (while merely causing ovarian failure in humans *Aittomäki et al., 1995*). Since many peptide hormones also affect gonadal development, it can therefore be challenging to explore both male and female physiology using these classical models.

Here, we use the naturally short-lived turquoise killifish, which exhibits one of the fastest recorded times to puberty among all vertebrate species (2–3 weeks, ~ sixfold faster than zebrafish and medaka). An additional and underexplored advantage of the killifish model is that, similar to mammals, sexual differentiation is germline-independent (*Harel et al., 2015*). This enables us to investigate the effect of a wide range of hormonal manipulations that can affect germline development on both sexes. Based on this mammalian-like trait, we hypothesize that hormonal perturbations in killifish might produce phenotypes that more faithfully recapitulate the corresponding mouse models, than do similar mutations observed in zebrafish (*Hu et al., 2019*; *Slanchev et al., 2005*; *Kurokawa et al., 2007*).

**Figure 1.** Perturbation of the killifish hypothalamic-pituitary-somatic axis. (**A**) Schematic illustration of the vertebrate hypothalamic-pituitary system, including members of the hypothalamic-pituitary-gonadal axis (HPG), the hypothalamic-pituitary-somatic axis (HPS), and the hypothalamic-pituitary-thyroid axis (HPT). These hormones are released from the pituitary, and travel through the bloodstream to bind/activate their target receptors/organs. FSH: follicle stimulating hormone; GH: growth hormone; TSH: thyroid stimulating hormone. (**B**) Top: generation of CRISPR mutants for *gh1*, with the guide RNA (gRNA) targets (red), protospacer adjacent motif (PAM, in bold), and indels. The mutation site is marked with a red asterisk. Bottom: Comparison of fish size between 8-week-old WT and $gh1^{\Delta4/\Delta4}$ male (top) and female (bottom) fish. Scale bar: 3.5 mm.

The online version of this article includes the following source data and figure supplement(s) for figure 1:

**Source data 1.** Physiological effect of growth hormone perturbations.

**Figure supplement 1.** Physiological effect of growth hormone perturbations.

**Figure supplement 1—source data 1.** Corresponding to *Figure 1—figure supplement 1A, B*.

Here we describe the effect on somatic growth and reproduction of perturbing three primary pituitary hormones (*Figure 1A*), namely growth hormone (*gh1*, the pituitary-somatic axis), follicle stimulating hormone (*fshb*, the pituitary-gonadal axis), and thyroid stimulating hormone (*tshb*, the pituitary-thyroid axis). The subsequent phenotypes indicate that although the killifish undergoes rapid growth and puberty, it still follows a vertebrate-conserved genetic program. This allows us to use our data to improve the understanding of how these hormones regulate the onset and duration of specific traits across evolutionary distances.

In addition, we report the use of a gain-of-function approach in which the investigated hormone is tagged using a self-cleavable fluorescent reporter. This vector is then ectopically expressed in vivo through intramuscular electroporation. Our results indicate that a single electroporation is enough to restore growth and reproductive phenotypes for several months. In addition, we describe the development of a doxycycline (Dox)-inducible system, which enables tunable expression patterns, and we demonstrate the feasibility of multiplexing interventions. Our platform therefore represents a simple and robust system for investigating the loss/gain-of-function of circulating factors in fish. Importantly, this strategy is highly effective after a single injection, and can readily be adapted to other fish species. Ultimately, a better understanding of how pituitary hormones shape the vertebrate life-history could allow us to uncouple specific traits, such as somatic growth or reproduction, from lifespan.

## Results

### Generation of a growth hormone CRISPR mutant in killifish

The hypothalamic–pituitary–somatic (HPS) axis involves the secretion of growth hormone (GH) from the pituitary gland and the consequent stimulation of insulin-like growth factor 1 (IGF-1) production in peripheral tissues (*Le Gac et al., 1993*; *List et al., 2011*; *Canosa et al., 2007*). We applied our recently developed CRISPR/Cas9 genome editing protocols (*Harel et al., 2015*; *Astre et al., 2022b*; *Astre et al., 2022a*; *Harel et al., 2016*) to perturb the HPS axis in the turquoise killifish by targeting exon 2 of the killifish *gh1* (*Figure 1B*, top). Crossing F0 founders with wild-type fish, allowed us to identify a 4 bp deletion in F1 fish that could cause a frameshift mutation. The *gh1$^{\Delta4/+}$* heterozygous mutants were viable and fecund, and were used to generate homozygous *gh1$^{\Delta4/\Delta4}$* mutants for phenotypic analysis.

### Mutating the killifish growth hormone delays somatic growth and maturation

Both male and female *gh1$^{\Delta4/\Delta4}$* homozygous mutants are strikingly smaller than control fish (*Figure 1B*, bottom). The proportion of *gh1$^{\Delta4/\Delta4}$* mutants, out of the total number of genotyped individuals, was roughly half of the expected 25:50:25 Mendelian distribution (*Figure 1—figure supplement 1A*). Most of this reduction can be attributed to their significantly smaller size, since smaller fish are more easily outcompeted in the communal tanks until individually housed following genotyping (at 4 weeks of age). No growth phenotype was detected in heterozygous fish (*Figure 1—figure supplement 1B*), possibly due to the over saturated GH expression required for rapid killifish growth.

Quantifying fish length, indicated that while both males and females are roughly 40% smaller than wild-type fish (*Figure 2A*), they remain fertile (*Figure 2B*). However, their slow growth is coupled with a significant delay in the onset of maturity (*Figure 2B*), and a proportional delay in peak fertility (*Figure 2C*, left). The maximal number of eggs at peak fertility is also reduced in mutant couples (*Figure 2C*, right). This difference in reproductive output is probably a result of the smaller size of the female body cavity and ovaries (*Figure 2—figure supplement 1A*), and the fact that egg size remains unchanged (*Figure 2—figure supplement 1B*).

To explore germline maturation at greater detail, we stained 2-month-old fish gonads with hematoxylin & eosin (H&E). Specifically, we examined whether GH deficiency causes changes in germ cell development, by quantifying the proportion of germ cells at each developmental stage in both sexes (*Figure 2D–F*). Interestingly, the results revealed that once the mutants reach maturity, oocyte, and sperm maturation are generally unaffected by the deficiency of growth hormone. Taken together, our findings demonstrate that like in mammalian models (*Zaczek et al., 2002*; *Danilovich et al., 1999*), GH deficiency delays both somatic growth and reproduction.

### Developing an efficient and scalable method to rescue hormonal perturbations

In order to rescue GH deficiency, we developed a gain-of-function system in which a hormone of interest can be ectopically expressed by optimizing intramuscular electroporation (*Figure 3A*, and see Materials and methods; *Callahan et al., 2018*; *Terova et al., 2013*; *Rao et al., 2008*; *Thummel et al., 2011*). *Gh1* from turquoise killifish cDNA was cloned upstream to a cassette encoding for GFP, and separated by the T2A self-cleaving peptide (*Liu et al., 2017*; *Figure 3A*, left). We injected the construct (~3 μg, in a volume of 3 μl) into the muscle of WT fish, and imaged GFP expression 72 hr after electroporation (*Figure 3A*, right). Testing injected but non-electroporated fish as controls further confirmed the contribution of the electroporation procedure (*Figure 3B*).

Since GH and GFP are co-translated, they should maintain a 1:1 ratio, and since the GFP tag is removed after the self-cleavage, hormonal secretion and function is expected to be unaffected. Thus, this approach enables faithful visualization of the electroporation efficiency, which serves as a proxy for hormone expression. GH expression, specifically in GFP-positive muscle fibers, could be directly detected by immunofluorescence in both WT fish (*Figure 3C*) and *gh1$^{\Delta4/\Delta4}$* mutants (*Figure 3—figure supplement 1A*).

As the next step, we electroporated the same construct into the muscle of male and female *gh1$^{\Delta4/\Delta4}$* mutant fish and monitored their somatic growth, with an empty vector containing *T2A-GFP*

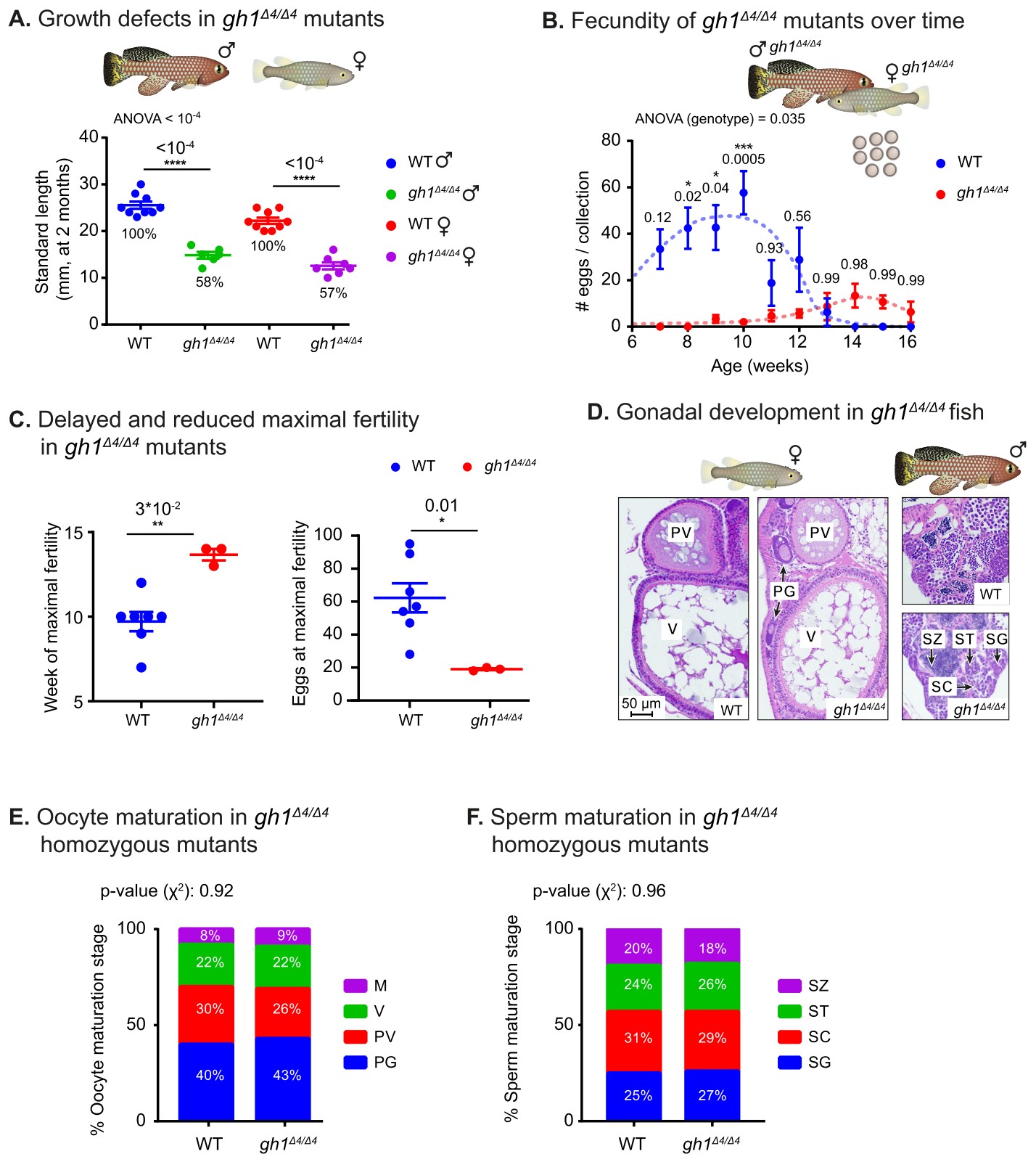

**Figure 2.** Phenotypic analysis of growth and reproduction in *gh1^Δ4/Δ4^* mutants. (**A**) Quantification of somatic growth (standard length) of 8-week-old WT or *gh1^Δ4/Δ4^* mutants: males (left) and females (right), n≥6 individuals from each experimental group. Error bars show mean ± SEM. Significance was calculated using one-way ANOVA with a Sidak post-hoc comparing the male and female mutants to the respective WT. Exact p-values are indicated. The relative size of the mutant fish, as compared to the corresponding controls, is indicated as %. (**B**) Quantification of reproductive output in *gh1^Δ4/^*

*Figure 2 continued on next page*

*Figure 2 continued*

Δ4 mutant pairs over time. Each dot represents the mean number of eggs of the indicated genotypes, per week of egg collection. There were three independent mutant pairs and seven WT pairs. Error bars show mean ± SEM. Significance was calculated using repeated measures two-way ANOVA with a Sidak post-hoc compared to the WT. Exact p-values are indicated. (**C**) Left: Quantification of the timing of peak fertility of $gh1^{Δ4/Δ4}$ mutant pairs. Each dot represents the age (in weeks) at which a breeding pair gave the maximal number of eggs. Right: reproductive output of $gh1^{Δ4/Δ4}$ mutant pairs at peak fertility. Each dot represents the maximal number of eggs per week of a single breeding pair under the indicated experimental conditions. This figure uses the data presented in **B**. Error bars show mean ± SEM. Significance was calculated using an unpaired Student's t-test. Exact p-values are indicated. (**D**) Representative histological sections, depicting ovaries and testis of the indicated genotypes. n≥4 individuals (two-month-old), from each genotype. Scale bar: 50 µm. PG: primary growth; PV: pre-vitellogenic; V: vitellogenic. SG: spermatogonia; SC: spermatocytes; ST: spermatids; SZ: spermatozoa. (**E**) Distribution of oocyte development stages. Data are presented as the proportion of each developmental stage of the indicated genotypes. n≥4 individuals for each experimental group. Significance was measured by $\chi^2$ test with the WT value as the expected model and FDR correction. Percentages and exact p-values are indicated. Oocyte developmental stages as previously reported (*Longenecker and Langston, 2016*) are indicated as follows: PG: primary growth; PV: pre-vitellogenic; V: vitellogenic; M: Mature. (**F**) Quantification of sperm maturation, examples in **D**. Data are presented as the proportion of each developmental stage of the indicated genotypes. n≥4 individuals for each experimental group. Significance was measured by $\chi^2$ test with the WT value as the expected model and FDR correction. Percentages and exact p-values are indicated. Sperm developmental stages as previously reported (*Longenecker and Langston, 2016*) are indicated as follows: SG: spermatogonia; SC: spermatocytes; ST: spermatids; SZ: spermatozoa.

The online version of this article includes the following source data and figure supplement(s) for figure 2:

**Source data 1.** Reproduction in WT fish.

**Source data 2.** Reproduction in $gh1^{Δ4/Δ4}$ mutants.

**Source data 3.** Corresponding to *Figure 2A–C, E and F* and *Figure 2—figure supplement 1B*.

**Figure supplement 1.** Effect of growth hormone deficiency on ovary and egg size.

**Figure supplement 1—source data 1.** Effect of growth hormone deficiency on ovary and egg size.

electroporated as a control. A moderate increase in average size was detected 3–4 weeks following electroporation (*Figure 3D*), with electroporated male and female mutants being ~10% larger than non-electroporated specimens (*Figure 3D*). Fish retain normal body dimensions (*Figure 3—figure supplement 1B*), suggesting uniform body-wide hormone effect. This partial phenotypic rescue was observed in both males and females, but was more pronounced in the larger males (*Figure 3D*).

The partial rescue in somatic growth might be due to the age at intervention. Specifically, electroporation was performed at 8 weeks of age, which is already past the period during which the fish exhibit a growth spurt and the onset of puberty. This timepoint was selected because of the difficulties in injection and electroporation in the extremely small young mutants. In contrast to the partial growth rescue, reproductive defects were fully rescued in GH-electroporated mutant couples (*Figure 3E*), as evidenced by an increase in the size of the ovary (*Figure 3F*). Together, these results demonstrate that ectopic muscle electroporation can rescue many aspects of GH deficiency.

## Reversible perturbation of the reproductive axis

We also mutated two additional representative pituitary hormones, namely follicle and thyroid stimulating hormones (*Figures 1A and 4A*). Disruption of thyroid hormone signaling produces both reproductive defects (lack of mature oocytes) as well as growth defects in both mice (*Tsujino et al., 2013*) and zebrafish (*Song et al., 2021*). Accordingly, homozygous $tshb^{Δ10/Δ10}$ killifish mutants display growth malformations (i.e. altered length/height ratio), as well as defects in pigmentation and in oocyte maturation (*Figure 4B and C*). These observations demonstrate that the pituitary-thyroid axis can be successfully modeled in killifish.

Finally, in order to investigate whether the extremely rapid sexual maturation in killifish (*Harel, 2022*; *Vrtílek et al., 2018*) follows vertebrate-conserved regulatory networks, we perturbed the pituitary-gonadal reproductive axis. Specifically, we imaged H&E stained tissue sections from $fshb^{in1/in1}$ mutants and WT females (n>4 for each experimental condition). Our findings demonstrated a striking reduction in mature oocytes in homozygous $fshb^{in1/in1}$ females (*Figure 4D*, representative image), indicating a role for FSHB in advanced stages of oogenesis.

The versatility of our gain-of-function approach, and the ability to restore mature oocytes in $fshb^{in1/in1}$ females was demonstrated by cloning *fshb* from turquoise killifish cDNA, and using a CMV:*fshb-T2A-GFP* plasmid to rescue mutant fish as described above (*Figure 4E*). Excitingly, while oocyte maturation and fertility were compromised in $fshb^{in1/in1}$ females, plasmid electroporation

**Figure 3.** Phenotypic rescue of growth hormone deficiency. (**A**) Left: schematic illustration of the gain-of-function plasmid, and intramuscular ectopic plasmid electroporation. Right: GFP is visible following the electroporation of a plasmid encoding for GH-T2A-GFP. (**B**) Representative images of the transgenic *Killibow* line (*Rozenberg et al., 2023a*) (expressing dTomato under the ubiquitin promoter, in red), injected with a plasmid expressing CMV:GFP (green), either without (top) or with (bottom) electroporation. n≥4 from each experimental group. (**C**) Immunofluorescence for GH expression

*Figure 3 continued on next page*

*Figure 3 continued*

(red) on muscle cryosections of electroporated fish. GFP expression is shown in green, and nuclear staining (DAPI) in blue. Representative image from each experimental group (n≥3). Scale bar: 50 μm. (**D**) Left: Quantification of somatic growth (standard length) of 12-week-old *gh1*$^{Δ4/Δ4}$ mutants following electroporation of a *gh1-T2A-GFP* plasmid as compared to WT control and controls injected with an empty vector: females (left) and males (center), n≥4 individuals from each experimental group. Error bars show mean ± SEM. Significance was calculated using one-way ANOVA with a Dunnet post-hoc compared to the WT. Exact p-values are indicated. The relative size of the mutant fish as compared to the corresponding controls is indicated as %. Right: Representative images of 12-week-old males (left) and females (right) from the indicated experimental groups. Scale bar: 3 mm. (**E**) Quantification of reproductive output in rescued *gh1*$^{Δ4/Δ4}$ mutant pairs. Each dot represents the number of eggs per indicated breeding pair, per week of egg collection. Error bars show mean ± SEM. n≥3 pairs for each experimental group. Error bars show mean ± SEM. Significance was calculated using one-way ANOVA with a Dunnet post-hoc compared to the WT. Exact p-values are indicated. (**F**) Representative images of ovaries from fish of the indicated experimental group. mature eggs are marked by a dashed circle. Scale bar: 1 mm.

The online version of this article includes the following source data and figure supplement(s) for figure 3:

**Source data 1.** Delivery of growth hormone via intramuscular electroporation.

**Source data 2.** Phenotypic rescue of growth hormone deficiency.

**Source data 3.** Corresponding to *Figure 3D, E* and *Figure 3—figure supplement 1B*.

**Figure supplement 1.** Delivery of growth hormone via intramuscular electroporation.

**Figure supplement 1—source data 1.** Intramuscular electroporation controls.

---

fully restored reproduction to WT levels (*Figure 4F*). The fertility of *fshb*$^{in1/+}$ females was unaffected (*Figure 4—figure supplement 1A*), suggesting that, as seen with GH signaling, reducing hormonal concentrations by half can still leave enough for phenotypic saturation.

To attempt to visualize circulating hormones, we removed the T2A linker from the electroporation plasmid, so that a FSHB-GFP fusion protein could be produced. However, while reproductive defects were fully restored following electroporation of the FSHB-GFP plasmid (*Figure 4—figure supplement 1B*), only faint GFP signals were detected in the gonads (*Figure 4—figure supplement 1C*), possibly due to a body-wide dilution, and the observation that very low concentrations of hormone are sufficient for functional stimulation.

## Developing a tunable expression system that is compatible with pulsatile release

In humans, the regulation of reproductive processes, including puberty and menstrual cycles, depends on precise temporal changes in hormone levels (*Taylor et al., 2019*). For example, gonadotropin-releasing hormone (GnRH) is required to stimulate the release of LH and FSH in prepubescent girls. At a maximum, the peak amplitude of LH increases about 10-fold, with only a doubling of the FSH pulse (*Taylor et al., 2019*) (see a schematic diagram in *Figure 5A*, top). After puberty, a specific sequence is also critical for ovulation (*Figure 5A*, bottom).

With the aim of achieving tunable control of hormonal levels, we electroporated a wide range (24 ng-3 μg) of plasmid concentrations. This produced a dose-dependent GFP expression (*Figure 5B*, top), which was relatively reproducible (*Figure 5C*). Injecting two plasmids encoding for hormones tagged by different fluorophores resulted in expression of both fluorophores indicating that the system is amenable to multiplexing (*Figure 5B*, bottom). Strikingly, monitoring *fshb*$^{in1/in1}$-rescued fish, indicated that a single injection can stably restore the physiological effect on fertility for at least 2 months, with a full effect achieved even at a relatively low plasmid concentration (*Figure 5D*).

While these experiments demonstrate the ability to control expression levels and perform multiplexed interventions, some expression patterns require delayed activation or reversible pulsations (*Figure 5A*). In order to address such issues, we introduced a Dox-inducible overexpression system (*Figure 5E*, top). This system includes a Dox-dependent transcriptional activator protein (rtTA) and a tet operator (tetO) containing promoter (TRE promoter) (*Das et al., 2016*) where the rtTA binds to the tetO promoter and activates gene expression only in the presence of doxycycline. Fish were electroporated with a tetOn:*fshb-T2A-GFP* plasmid we constructed and, 3 days later, were exposed to Dox for 72 hr. This resulted in a GFP signal (*Figure 5d*, bottom), thereby confirming the applicability of this approach in-vivo.

Our Dox system was also used for functional interventions. Specifically, we electroporated the tetOn:*fshb-T2A-GFP* plasmid into *fshb*$^{in1/in1}$ mutant females, which were then exposed to Dox for 72 hr

## A. Generating a CRISPR/Cas9 mutant for the killifish *tshb* and *fshb* genes

*fshb* (XM_015953905.1)

*fshb$^{in1}$* (exon 2)     0.2 kb

ACCAGTCCCCGAATACAGGTGTTCACCTCAGGCTG
ACCAGT**CCCC**G-**TTACAGGTGTTCACCTCA**GGCTG

*tshb* (XM_015958655.1)

*tshb$^{Δ10}$* (exon 2)     0.2 kb

GGGTGGA----------GTCATCCTACCCGGCTGT
GGGTGGAGTA**CCG**CACCGTCATCCTACCCGGCT**GT

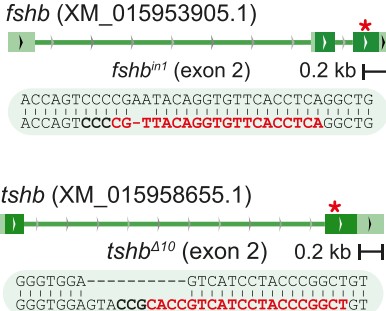

## B. Somatic growth and reproductive defects in *tshb* mutants

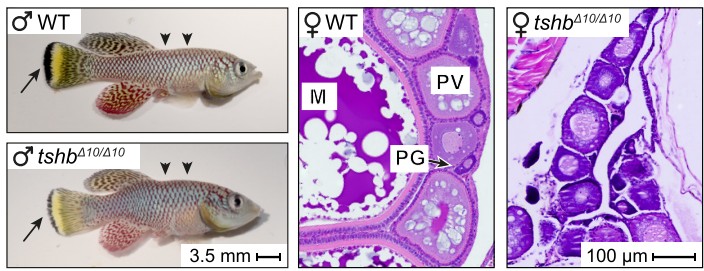

## C. Altered body proportion of *tshb$^{Δ10/Δ10}$* males

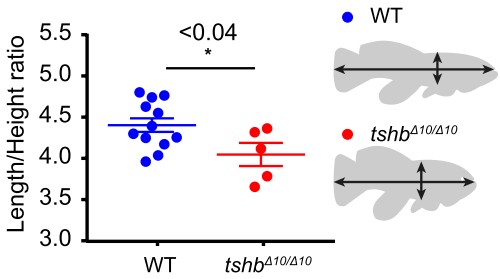

## D. Lack of mature oocytes in *fshb$^{in1/in1}$* females

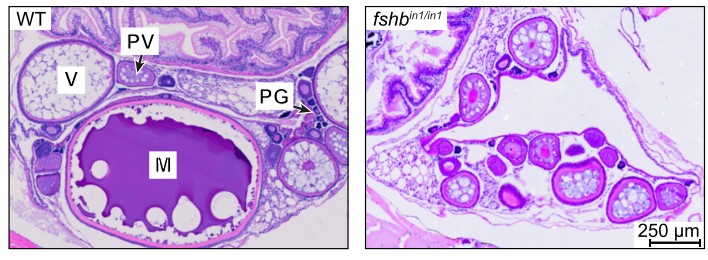

## E. Ectopic expression of FSHB can rescue ovarian phenotype

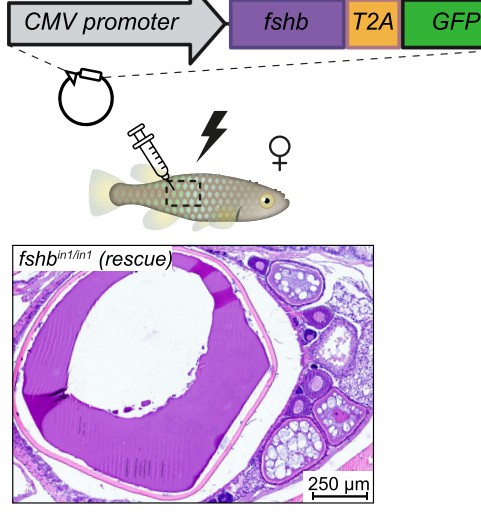

CMV promoter — *fshb* — T2A — GFP

*fshb$^{in1/in1}$* (rescue)     250 µm

## F. FSHB can rescue oocyte maturation and fertility defects in *fshb$^{in1/in1}$* females

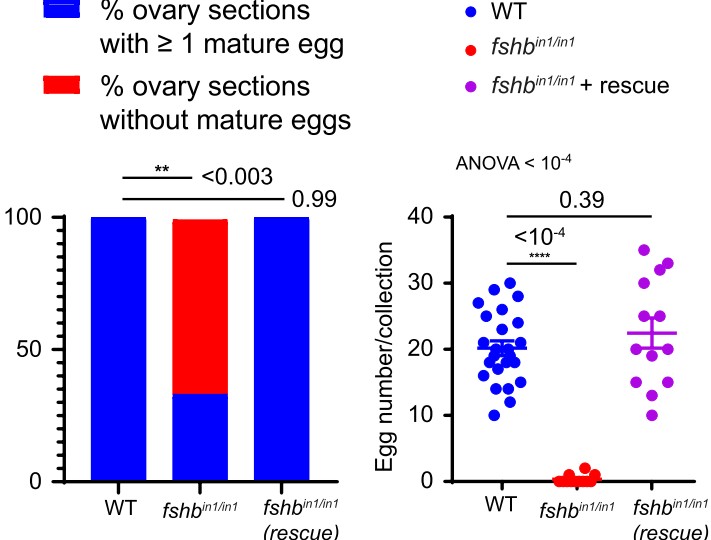

■ % ovary sections with ≥ 1 mature egg
■ % ovary sections without mature eggs

● WT
● *fshb$^{in1/in1}$*
● *fshb$^{in1/in1}$* + rescue

**Figure 4.** Reversible perturbation of the killifish reproductive axis. (**A**) Generation of CRISPR mutants for *fshb* and *tshb*, depicting the guide RNA (gRNA) targets (red), protospacer adjacent motif (PAM, in bold), and indels. Red asterisk marks the targeted exon. (**B**) Left: comparison of 2-month-old WT (top) and *tshb$^{Δ10/Δ10}$* male fish (bottom). Black arrows highlight tail melanocytes, while arrowheads indicate alterations in body shape. Right: representative histological sections demonstrating that 1-month-old *tshb$^{Δ10/Δ10}$* fish (right) lack mature oocytes compared to WT ovaries (left). Oocyte developmental

*Figure 4 continued on next page*

*Figure 4 continued*

stages as reported previously (*Longenecker and Langston, 2016*) are: PG: primary growth; PV: pre-vitellogenic; V: vitellogenic; M: Mature. Scale bar: 100 μm. (**C**) Left: Quantification of the ratio between length and height of experimental fish as indicated. Significance was calculated using an unpaired Student's t-test. Exact p-values are indicated. Right: a schematic model of the measurements used. (**D**) Representative histological sections, depicting 1-month-old ovaries of the indicated genotypes. n≥4 individuals from each genotype. Scale bar: 250 μm. Oocyte developmental stages as reported previously (*Longenecker and Langston, 2016*) are: PG: primary growth; PV: pre-vitellogenic; V: vitellogenic; M: Mature. (**E**) Representative histological sections of ovaries in one-month-old *fshb*^in1/in1^ mutant females, one week following electroporation of an *fshb-T2A-GFP* plasmid. Representative of n≥3 individuals Oocyte developmental stages as reported previously (*Longenecker and Langston, 2016*) are: PG: primary growth; PV: pre-vitellogenic; V: vitellogenic; M: Mature. Scale bar: 250 μm (**F**) Left: proportion of histological sections in which at least one mature egg has been detected. Significance was calculated using Fisher's exact test with an FDR correction, Exact P-values are indicated. Right: quantification of female fertility. Each dot represents the number of eggs per indicated breeding pair, per week of egg collection. The data are from at least 3 independent pairs and 4 independent collections. Error bars show mean ± SEM with individual points. Significance was calculated using one-way ANOVA with a Dunnet post-hoc compared to the WT and exact p-value is indicated.

The online version of this article includes the following source data and figure supplement(s) for figure 4:

**Source data 1.** Reversible perturbation of the killifish reproductive axis.

**Source data 2.** Corresponding to *Figure 4C and F* and *Figure 4—figure supplement 1A, B*.

**Figure supplement 1.** Physiological effects of FSHB levels.

and monitored weekly for changes in reproductive output. Interestingly, fertility defects were fully rescued after 2–3 weeks, but only within a narrow temporal window (*Figure 5F*). Electroporation of the plasmid without subsequent exposure to Dox was sufficient to produce a minor effect (*Figure 5F*), which could be attributed to promoter leakiness. This demonstrates that extremely low hormone levels are required to reach the threshold of a functional rescue. Thus, our findings indicate that this approach can successfully produce time- and dose-dependent hormone expression.

## Discussion

Here, we describe the development of a platform that can rapidly and reversibly manipulate life history traits in fish. Specifically, we genetically perturb pituitary hormones, and rescue hormone loss-of-function by ectopically expressing the missing hormones after intramuscular electroporation. This methodology can be used to demonstrate that the explosive growth and rapid puberty onset in killifish are regulated by vertebrate-conserved paradigms. Significantly, the naturally compressed killifish life cycle shortens experimental timescales, and allows for a quick physiological readout of both loss- and gain-of function. Together, this method is relatively high-throughput, and facilitates large-scale interrogation of peptide hormones in fish.

Each hormonal manipulation produced distinct and overlapping effects on somatic growth and reproduction. This apparent co-regulation is in accordance with several evolutionary theories that predict functional trade-offs between life-history traits (*Austad and Hoffman, 2018*; *Maklakov and Chapman, 2019*; *Kirkwood and Holliday, 1979*; *Kirkwood and Austad, 2000*). Similarly, pituitary hormones also link somatic growth with organismal lifespan in the long-lived Ames and Snell dwarf mice, which suffer from a deficiency of pituitary growth hormone (among other hormones; *Flurkey et al., 2002*; *Flurkey et al., 2001*; *Bartke and Brown-Borg, 2004*). Mice with a mutated growth hormone receptor are long-lived (*List et al., 2011*), and in humans, longer lives and cancer protection are observed in Laron Syndrome patients (dwarfism due to a growth hormone receptor mutation; *Laron et al., 2017*). These complex relationships raise interesting predictions. For example, could a specific delay in maturity drive proportional changes in vertebrate lifespan? A better understanding of these mechanisms will allow us to uncouple life-history traits, such as somatic growth and reproduction (*Figure 6*).

Our hormonal perturbations highlight several interesting aspects in killifish. One observation is that a very low level of FSHB is required to support normal reproduction. This might be linked to the extremely rapid maturation process and asynchronous cycle in killifish, which might require high sensitivity. In addition, the delay between Dox treatment and the maximal effect on reproduction (*Figure 5F*), is probably linked to the length of the natural egg maturation process in killifish. The question of whether rescue of growth prior to the onset of puberty can indeed lead to a more pronounced phenotype will require further investigation.

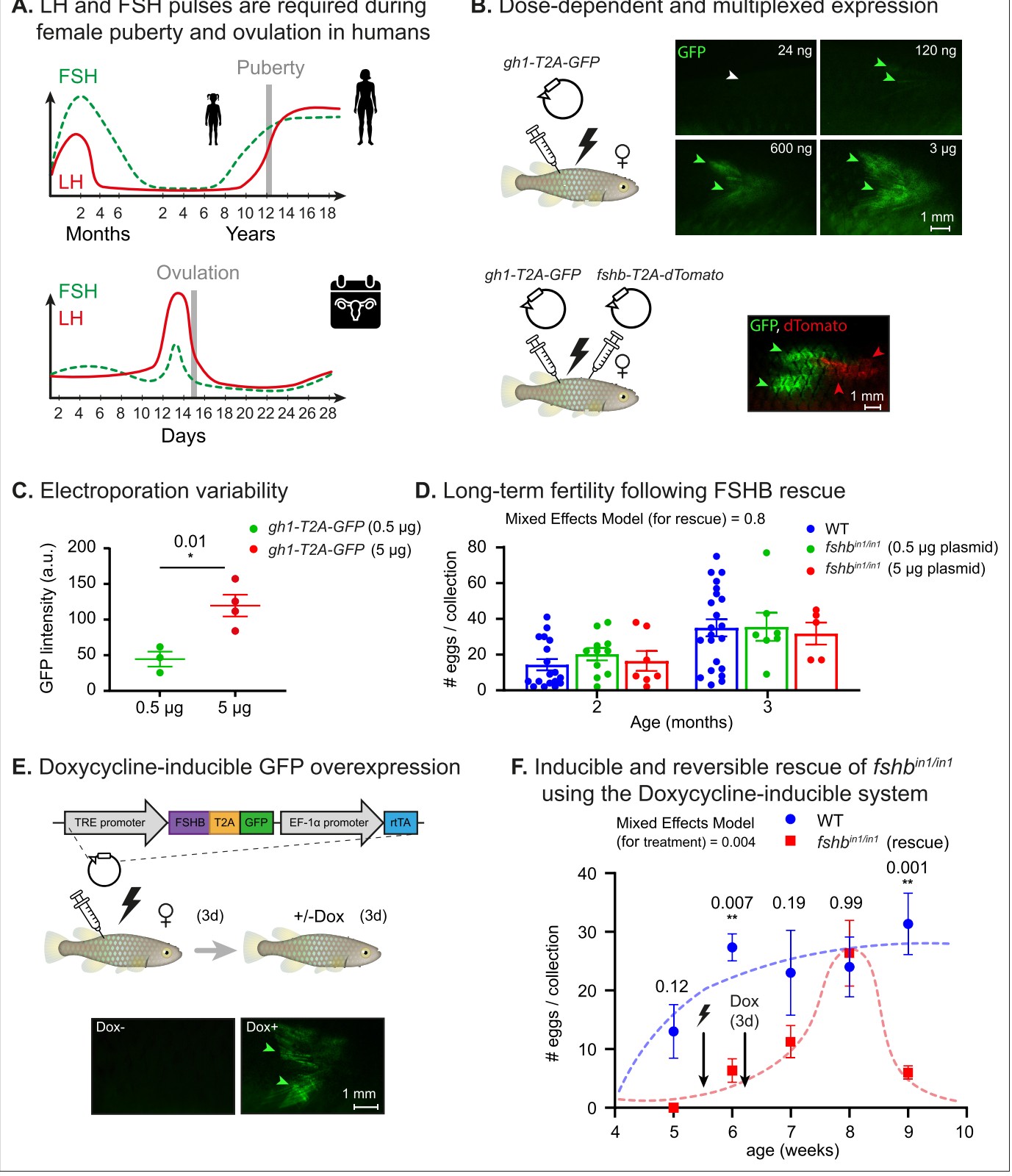

**Figure 5.** Dose dependent, multiplexed, and inducible expression systems. (**A**) A schematic illustration depicting complex expression levels of LH (red line) and FSH (green line) during puberty (top) and ovulation (bottom) in human females. Adapted from ***Taylor et al., 2019***. (**B**) Top: electroporation of the indicated plasmid concentrations. GFP signal (green arrowheads) and lack of signal (white arrowheads) are shown. n≥3 for each experimental condition. Scale bar = 1 mm. Bottom: electroporation of the indicated plasmids. GFP signal (green arrowheads) and dTomato signal (red arrowheads)

*Figure 5 continued on next page*

*Figure 5 continued*

are shown. Scale bar = 1 mm. (**C**) Quantification of GFP intensity (arbitrary units) in muscle of fish injected with 500 ng or 5 µg of a plasmid encoding for CMV:GFP. n≥3 for each experimental condition. Error bars show mean ± SEM. Significance was calculated using an unpaired Student's t-test and the exact p-value is indicated. (**D**) Right: Quantification of fertility over time of *fshb*[in/in1] fish rescued with a plasmid expressing FSHB. Each dot represents the number of eggs per indicated breeding pair, per week of egg collection. The data are from at least four independent pairs and four independent collections each month. Error bars show mean ± SEM with individual points. Significance was calculated using a Mixed Effects Model. Exact p-values are indicated.(**E**) Top: schematic illustration showing electroporation of a plasmid with a doxycycline-inducible promoter, coding for a desired protein (ProteinX) and tagged with a T2A-GFP. Bottom: a GFP signal is observed (green arrowheads) following Dox treatment. In control fish (-Dox), there is no detectable signal (white arrowheads). n≥3 for each experimental condition. Scale bar = 1 mm. (**F**) Quantification of reproductive output over time in *fshb*[in/in1] females rescued with a plasmid expressing FSHB under Dox induction. Each dot represents the mean number of eggs of the indicated experimental group, per week of egg collection. n≥3 for each experimental condition. Timing of electroporation and Dox treatment is indicated. Error bars show mean ± SEM. Significance was calculated using a Mixed-Effects Model. Exact p-values are indicated.

The online version of this article includes the following source data for figure 5:

**Source data 1.** Dose dependent, multiplexed, and inducible expression systems.

**Source data 2.** Corresponding to *Figure 5C, D and F*.

Notably, some hormonal perturbations in killifish mutants seem to recapitulate mammalian models (*Zaczek et al., 2002*; *Danilovich et al., 1999*) more faithfully than similar alterations in zebrafish (*Hu et al., 2019*). For example, while zebrafish *gh* mutants exhibit an 80% death rate, possibly due to gut malformations, the lower death rate seen in killifish is apparently due only to housing conditions, and the fish display relatively normal gut architecture (*Figure 2—figure supplement 1A*). In addition, while zebrafish *gh* mutants display an arrest of female maturity, with only a few mutant females undergoing puberty, fertility is merely delayed in killifish HPS axis mutants and fecundity is reduced due to spatial constraints, while all germline developmental stages are normally present (*Figure 2D–F*, and *Figure 2—figure supplement 1A*). In contrast to mice (*Rowland et al., 2005*) and zebrafish (*Hu*

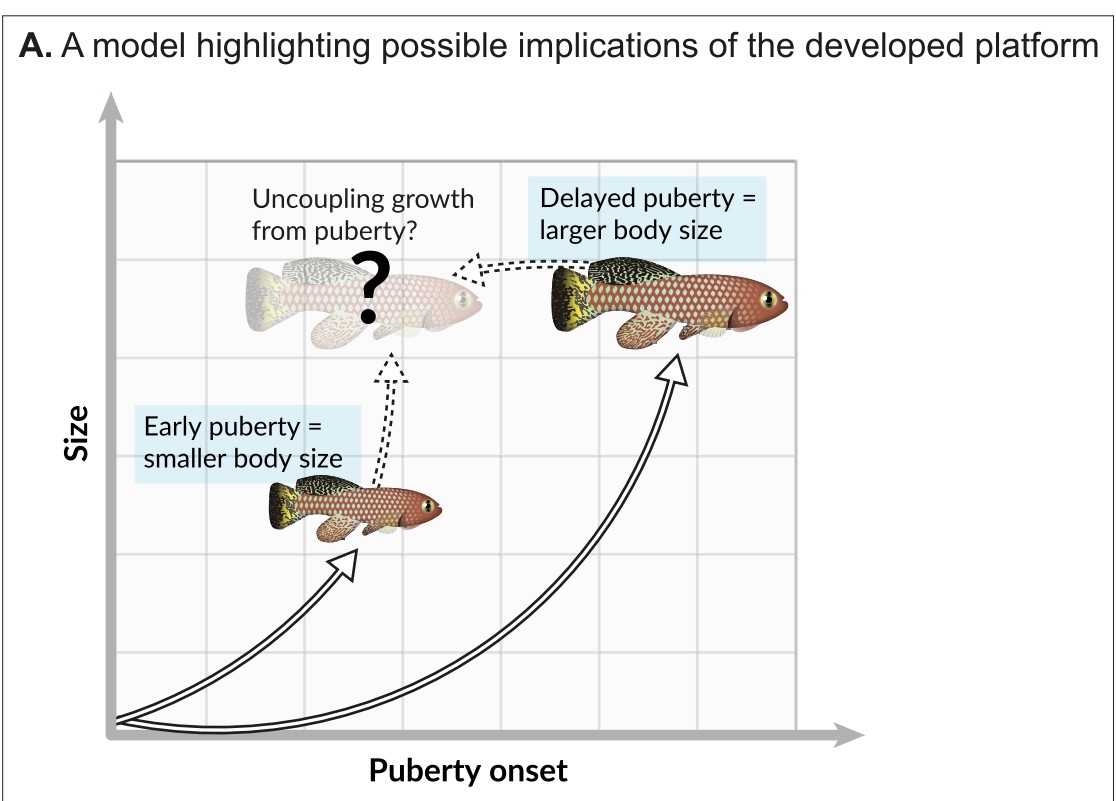

**Figure 6.** Possible implications. (**A**) Onset of maturity can negatively affect growth. Identifying the molecular mechanisms that regulate these seemingly opposing traits, and potentially uncoupling them, holds great promise for basic research and aquaculture.

*et al., 2019*), a heterozygous *gh1* mutation does not have an obvious effect on the growth of killifish. This phenomenon may be due to the explosive growth killifish exhibit, or to the possibility that heterozygotes killifish are merely able to 'catch up' (similar to zebrafish *Hu et al., 2019*).

So far, we and others have primarily perturbed one hormone at a time. However, in the real world, multiple signals are integrated to determine the duration and onset of life-history traits. Importantly, our method is compatible with multiplexing (*Figure 5B*), and could allow an investigation of the combinatorial effects of several hormones. Similarly, the dose-dependent and doxycycline-inducible system can be used to provide information about the temporal windows required to achieve a sequential or reversible gain-of-function.

We predict that our approach will be of great use in the optimization of commercially valuable traits in aquaculture (*Taranger et al., 2010*). For example, current manipulations in certain eel species are extremely time consuming and require ~16 weekly injections of pituitary extracts to induce maturation (*Okamura et al., 2014*). Our system, which has a long-lasting effect without modifying the host genome, will allow simple plasmid electroporation to replace traditional methods of hormone administration for the purposes of manipulation of the reproductive cycle, growth, and resistance to disease.

We have recently demonstrated that the killifish model can be used to identify novel regulators of aging, which involve systemic modulation of metabolism (*Astre et al., 2022a*). Therefore, it should be possible to utilize this platform to investigate systemic factors as modifiers of health and longevity. For example, recent studies have demonstrated that circulating factors in young plasma can rejuvenate old mice (*Baht et al., 2015*; *Conboy et al., 2005*; *Huang et al., 2018*; *Katsimpardi et al., 2014*; *Loffredo et al., 2013*; *Salpeter et al., 2013*; *Sinha et al., 2014*; *Villeda et al., 2011*; *Villeda et al., 2014*; *Grunewald et al., 2021*). So far, only a handful of these factors, including VEGF (*Grunewald et al., 2021*) (Vascular Endothelial Growth Factor), and GDF11 (*Sinha et al., 2014*) (Growth Differentiation Factor 11) have been experimentally tested, and there are many possible candidates for age-dependent changes in the plasma proteome that remain to be explored (*Lehallier et al., 2019*). In conclusion, our platform significantly advances the state-of the-art by providing easy and efficient tools that can be used to dissect the mechanisms that regulate vertebrate life history at an unprecedented resolution.

# Materials and methods

## Key resources table

| Reagent type (species) or resource | Designation | Source or reference | Identifiers | Additional information |
|---|---|---|---|---|
| Gene (*N. furzeri*) | *gh1* | NCBI | Gene ID: 107390293 | |
| Gene (*N. furzeri*) | *fshb* | NCBI | Gene ID: 107381959 | |
| Gene (*N. furzeri*) | tshb | NCBI | Gene ID: 129164764 | |
| Genetic reagent (*N. furzeri*) | *gh1*$^{\Delta4}$ | This paper | | Line maintained at the Harel lab |
| Genetic reagent (*N. furzeri*) | *fshb*$^{in1}$ | This paper | | Line maintained at the Harel lab |
| Genetic reagent (*N. furzeri*) | *tshb*$^{\Delta10}$ | This paper | | Line maintained at the Harel lab |
| Recombinant DNA reagent | *CMV:gh1-T2A-GFP* | This paper, Addgene | 194883 | |
| Recombinant DNA reagent | *CMV:fshb-T2A-GFP* | This paper, Addgene | 194356 | |
| Recombinant DNA reagent | *Tol2-TLCV2* | This paper, Addgene | 196331 | |
| Recombinant DNA reagent | *TetOn:fshb-T2A-GFP* | This paper, Addgene | 205595 | |

*Continued on next page*

*Continued*

| Reagent type (species) or resource | Designation | Source or reference | Identifiers | Additional information |
|---|---|---|---|---|
| Recombinant DNA reagent | *fshb-GFP fused* | This paper, Addgene | 205596 | |
| Recombinant DNA reagent | *fshb-t2a-dTOMATO* | This paper, Addgene | 205597 | |
| Antibody | Anti rabbit polyclonal anti-tilpia-GH | Levavi-Sivan lab | | (1:100) |
| Sequence-based reagent | fshb sequencing forward | This paper | | GAGATCGCGGGCATGAACT |
| Sequence-based reagent | fshb sequencing reverse | This paper | | ACCACACTCATCCACACCAC |
| Sequence-based reagent | gh1 sequencing forward | This paper | | TAACCCTAGCCCATGTCGGT |
| Sequence-based reagent | gh1 sequencing reverse | This paper | | TTTTGTTGAGCTGACGCTGC |
| Sequence-based reagent | tshb sequencing forward | This paper | | TTGGAGTAAACAGGACAGCCG |
| Sequence-based reagent | tshb sequencing reverse | This paper | | TTCCCCGTGTGTCATTCAGG |
| Sequence-based reagent | fshb cloning forward | This paper | | ATGCAACTGGTTGTCATGGCAGC |
| Sequence-based reagent | fshb cloning reverse | This paper | | ACAGCCGAGTACGTGTGGATGGAAGG |
| Sequence-based reagent | gh1 cloning forward | This paper | | ATGGACAGAGCCCTCCTCCTCC |
| Sequence-based reagent | gh1 cloning reverse | This paper | | CAGAGTGCAGTTTGCTTCTGGA |

## Experimental model and subject details

### Data analysis

No data was excluded during the analysis, as no significant outliers or classical exclusion criteria (e.g. unnatural death) occurred during experimentation. We assumed normality for all data, as commonly applied for physiological traits (such as size, fertility etc.). Therefore, a Student's t-test could be used for comparing two groups, and ANOVA for more than two groups. In case of repeated measures over time, we used a two-way repeated measures ANOVA test with time as one variable. To calculate proportions within a group we used a $\chi^2$ test or Fisher's exact test. For biological replicates, we used parallel measurements of individual fish that captures random variation. Technical replicates were considered when the same experiment was conducted several times, such as several egg collections from the mating pair. Power analysis was performed for growth measurements, predicting a reduction of 50% in size with an alpha of 0.05 and power of 80%:

$$k = \frac{n_2}{n_1} = 1$$

$$n_1 = \frac{\left(\sigma_1^2 + \sigma_2^2/k\right)\left(z_{1-\frac{\alpha}{2}} + z_{1-\beta}\right)^2}{\Delta^2}$$

$$n_1 = \frac{\left(5^2 + \frac{5^2}{1}\right)(1.96 + 0.84)^2}{12.5^2}$$

$$n_1 = 3$$

$$n_2 = kn_1 = 3$$

### African turquoise killifish strain, husbandry, and maintenance

The African turquoise killifish (GRZ strain) was housed as previously described (*Astre et al., 2022b*; *Harel et al., 2016*). Fish were grown at the Hebrew University of Jerusalem (Aquazone ltd, Israel) in a central filtration recirculating system at 28 °C, with a 12 hr light/dark cycle. Fish were fed with live Artemia until the age of 2 weeks (#109448, Primo), and starting week 3, fish were fed three times a day

on weekdays (and once a day on weekends), with GEMMA Micro 500 Fish Diet (Skretting Zebrafish, USA), supplemented with Artemia once a day. All genetic models (described below) were maintained as heterozygous and propagated by crossing with wild-type fish. All turquoise killifish care and uses were approved by the Subcommittee on Research Animal Care at the Hebrew University of Jerusalem (IACUC protocols #NS-18-15397-2 and #NS-22-16915-3).

## CRISPR/Cas9 target prediction and gRNA synthesis

CRISPR/Cas9 genome-editing protocols were performed as described previously (*Astre et al., 2022b*). Briefly, evolutionary conserved regions upstream of functional or active protein domains were selected for targeting the selected genes. gRNA target sites were identified using CHOPCHOP (https://chopchop.rc.fas.harvard.edu/) (*Labun et al., 2019*), and are shown below. PAM sites are shown in bold, and when needed, the first base-pair was changed to a 'G' to comply with the T7 promoter.

| Gene name | | gRNA sequence | Exon number |
|---|---|---|---|
| *gh1* | (XM_015966915.1) | [G/A]GAAGAGTCTTTGAGCGAGC**AGG** | 2 |
| *fshb* | (XM_015953905.1) | [G/T]GAGGTGAACACCTGTAACG**GGG** | 2 |
| *tshb* | (XM_015958655.1) | [G/A]GCCGGGTAGGATGACGGTG**CGG** | 2 |

Design of variable oligonucleotides, and hybridization with a universal reverse oligonucleotide was performed according to *Astre et al., 2022b*, and the resulting products were used as a template for in vitro transcription. gRNAs were in vitro transcribed and purified using a quarter reaction of TranscriptAid T7 High Yield Transcription Kit (Thermo Scientific #K0441), according to the manufacturer's protocol.

## Production of Cas9 mRNA

Experiments were performed as described previously (*Harel et al., 2015*; *Astre et al., 2022b*). The pCS2-nCas9n expression vector was used to produce Cas9 mRNA (Addgene, #47929; *Jao et al., 2013*). Capped and polyadenylated Cas9 mRNA was in vitro transcribed and purified using the mMESSAGE mMACHINE SP6 ULTRA (Thermo Fisher # AM1340).

## Microinjection of turquoise killifish embryos and generation of mutant fish using CRISPR/Cas9

Microinjection of turquoise killifish embryos was performed according to *Astre et al., 2022b*. Briefly, nCas9n-encoding mRNA (300 ng/µL) and gRNA (30 ng/µL) were mixed with phenol-red (P0290, Sigma-Aldrich) and co-injected into one-cell stage fish embryos. Sanger DNA sequencing was used for detecting successful germline transmission on F1 embryos. Fish with desired alleles were outcrossed further to minimize potential off-target effects, and maintained as stable lines by genotyping using the KASP genotyping platform (Biosearch Technologies) with custom made primers. All primers used in generating the mutations can be found in the **Key Resources Table**.

## Method details

### Growth measurements

Both sexes were measured by imaging at the indicated timepoints with a Canon Digital camera EOS 250D, prime lens Canon EF 40 mm f/2.8 STM. To limit vertical movement during imaging, fish were placed in a tank with 3 cm deep water, and images were taken from the top using fixed lighting and height. A ruler was included in each image to provide an accurate scale. Body standard length was measured from the tip of the snout to the posterior end of the last vertebra (excluding the length of the tail fin). Height of the fish was measured from the posterior base of the dorsal fin to the posterior base of the anal fin. Measurements were preformed using Matlab (R2021a), by converting pixel number to centimeters using the included reference ruler. All fish measured were siblings, and for blinding, genotypes were determined after the experiments.

## Fertility analysis

Fish fertility was evaluated as described previously (*Harel et al., 2015*; *Astre et al., 2022a*). Briefly, three to seven independent age-matched pairs of fish (one male, one female) of the indicated genotypes were placed in the same tank. All breeding pairs were allowed to continuously breed on sand trays, and embryos were collected and counted on a weekly basis. Results were expressed as the number of eggs per couple per week of egg-lay. Significance compared to the WT was calculated using repeated measures two-way ANOVA with a Sidak post-hoc.

## Histology

### Hematoxylin and eosin

Tissues samples were processed as described previously (*Harel et al., 2015*; *Astre et al., 2022b*; *Harel and Brunet, 2015*; *Astre et al., 2021*; *Harel et al., 2022*; *Nathan et al., 2008*; *Theis et al., 2010*; *Harel and Tzahor, 2012*; *Harel et al., 2012*; *Benayoun et al., 2019*; *Rozenberg et al., 2023b*; *Astre et al., 2022a*; *Harel et al., 2016*; *Rozenberg et al., 2023a*; *Van Keymeulen et al., 2009*; *Moses and Harel, 2023*; *Valenzano et al., 2015*; *Harel et al., 2009*; *Gruenbaum-Cohen et al., 2012*). Briefly, fish were euthanized with 500 mg/l tricaine (MS222, #A5040, Sigma). Paraffin sections were prepared by opening the body cavity of the fish and following a 72 hr fixation in 4% PFA solution at 4 °C, samples were dehydrated and embedded in paraffin using standard procedures. Sections of 5–10 µm were stained with Hematoxylin and Eosin, and examined by microscopy. A fully motorized Olympus IX23 microscope with an Olympus DP28 camera was used to collect images. Stages of oocyte and spermatogenic cell development were identified as described previously (*Longenecker and Langston, 2016*).

### Immunohistochemistry

Fish were euthanized and dissected as previously described (*Astre et al., 2022b*). Muscle tissue was fixed in 4% PFA at 4 °C for 2 hr, and immersed in an OCT- sucrose solution for 2 hr at 4 °C. The OCT-sucrose solution is composed of 20% sucrose w/v (Bio-Lab #001922059100) and 30% OCT v/v (Scigen Scientific Gardena #4586) in PBS. Tissues were then transferred to OCT (12 h at 4 °C) and frozen in liquid nitrogen. All immersions were performed with mild shaking. Serial 20 µm sections were taken using a cryostat, airdried, and stored at –20 °C. For immunostaining, slides were washed 3 X in PBS, and permeabilized for 10 min in a permeabilization buffer containing 0.1% Triton (Avanator Performance materials #X198-07) and 1% BSA (Sigma-Aldritch #A7906 in PBS). Sections were then blocked for 10 min (DAKO #X0909), and incubated with primary antibodies overnight. The following primary antibody was used: rabbit anti-Nile Tilapia GH antibody (1:100), a generous gift from Prof. Berta Levavi-Sivan. After several washes, the sections were incubated for 1 hr at room temperature with donkey anti rabbit Alexa Fluor 594 secondary antibody (Abcam #150064, 1:500) in antibody diluent (DAKO #S0809). After several washes, autofluorescence was quenched using TrueVIEW autofluorescence quenching kit (Vector Labs #SP8500) according to the manufacturer's protocol and mounted with VECTASHIELD containing DAPI (Vector Labs #30326). Samples were imaged using a fully motorized Olympus IX23 microscope with a Photometrics BSI camera, and processed in imageJ (*Abràmoff et al., 2004*).

## Injection and ectopic over-expression of plasmids via electroporation

### Cloning of killifish cDNAs

Killifish cDNAs were cloned from brain tissues from male and female killifish by homogenization in TRIreagent (Sigma #T9424) using 3 mm Nirosta disruption beads (PALBOREG FEDERAL #BL6693003000) and a tissue homogenizer (TissueLyser LT, Qiagen #85600). Total RNA was isolated from the lysed tissues using the Direct-zol RNA miniprep kit. (Zymo research #R2052), and the Verso cDNA Synthesis Kit (Thermo scientific #AB1453A) was used to prepare cDNA with random primers according to the manufacturer's protocol. cDNA for the *gh1* and *fshb* was amplified using custom DNA oligonucleotides (Sigma) and Platinum SuperFi II DNA Polymerase (Invitrogen, #12361010). Primer sequences are available in the **Key Resources Table**. PCR products were purified (QIAquick PCR purification kit, Qiagen #28104) and sequence-verified. The sequence-verified ORFs were cloned using GIBSON (NEB, #E2611L) into the pLV-EGFP plasmid or our Dox inducible plasmid (#196331),

which was modified such that each hormone is tagged with a GFP, separated by the T2A self-cleaving peptide (*Liu et al., 2017*). Plasmids and corresponding annotated maps are available via Addgene (#194356, #194883, #196331, # 205595, # 205596, #205597).

## In-vivo electroporation

The electroporation protocol was adapted from *Callahan et al., 2018*. Fish were sedated in MS222 (200 mg/l), and 3–5 µl plasmid solution (Plasmid concentration200-1000 ng/µl) containing Phenol Red for visualization (0.1% Sigma #P0290) was injected intramuscularly using a Nanofil syringe (WPI, #NANOFIL). Fish were then electroporated using 7 mm tweezer electrodes (NEP GENE #CUY650P10) with the ECM 830 generator. (BTX #45–0661). Electrodes were coated with wet cotton to minimize the risk to the fish. Fish were electroporated with 6 pulses of 28V (14 V for small fish, such as $gh1^{\Delta4/\Delta4}$), for 60ms each with an interval of 1 s between each pulse. Bright field and florescent Images were recorded by a Leica MC190HD camera mounted on a Leica M156FC microscope. Variability of GFP intensity was calculated using ImageJ by measuring the 'integrated density' of the electroporated area and subtracting the area of electroporation times the mean intensity of the background (measured at three different points and then averaged).

## Dox inducible expression

Fish were injected with a modified version of the TLCV2 plasmid (Addgene #87360). This plasmid contains Dox induced Cas-9/GFP, which we modified to include Tol2 sites (Addgene #196331). Finally, we created a plasmid emcoding a Dox induced *fshb-T2A-GFP* (Addgene #205595) . Electroporation was performed as described above. Doxycycline treatment protocol was adapted from *Campbell et al., 2012*; *West et al., 2014*. Briefly, Doxycycline hyclate (Dox, Sigma-Aldrich #D9891) was dissolved in 100% ethanol at 10 mg/mL for storage, and was added to fish water for a final concentration of 10 µg/mL. Fish were treated for 72 hr, and fresh water with Dox was changed every 24 hr. Due to possible light sensitivity of the drug, tanks were protected from light during treatment. Immediately following the treatment fish were imaged using a Leica MC190HD camera mounted on a Leica M156FC microscope.

## Acknowledgements

We thank the Harel lab for stimulating discussions and feedback on the manuscript. We thank Ariel Velan, Ella Yanay, Ashayma Abu-tair, Yonatan Birenbaum, and Fatma Idrees for help with killifish maintenance and Adi Oron-Gottesman for her help with cloning. Supported by the Zuckerman Program (IH), ISF 2178/19 (IH), Israeli Ministry of Science 3–17631 (IH), 3–16872 (IH), the Moore Foundation GBMF9341 (IH), BSF-NSF 2020611 (IH), ERC StG #101078188 (IH), the Israeli Ministry of Agriculture 12-16-0010 (IH), the Levi Eshkol scholarship of the Israeli Ministry of Science (EM), the Czech Science Foundation (#22-01781O), and the Ministry of Education, Youth and Sports of the Czech Republic (#CZ.02.1.01/0.0/0.0/16_025/0007370) (RF).

# Additional information

## Funding

| Funder | Grant reference number | Author |
|---|---|---|
| Zuckerman Program | | Itamar Harel |
| Israel Science Foundation | 2178/19 | Itamar Harel |
| Israeli Ministry of Science | 3-17631 | Itamar Harel |
| Israeli Ministry of Science | 3-16872 | Itamar Harel |
| Moore Family Foundation | GBMF9341 | Itamar Harel |
| BSF-NSF | 2020611 | Itamar Harel |

| Funder | Grant reference number | Author |
| --- | --- | --- |
| European Research Council | StG #101078188 | Itamar Harel |
| Israeli Ministry of Agriculture | 12-16-0010 | Itamar Harel |
| Israeli Ministry of Science | Levi Eskol Fellowship | Eitan Moses |
| Czech Science Foundation | #22-01781O | Roman Franek |
| Ministry of Education, Youth and Sports of the Czech Republic | #CZ.02.1.01/0.0/0.0/16_025 /0007370 | Roman Franek |

The funders had no role in study design, data collection and interpretation, or the decision to submit the work for publication.

## Author contributions

Eitan Moses, Conceptualization, Resources, Data curation, Formal analysis, Validation, Investigation, Visualization, Methodology, Writing – original draft, Project administration, Writing – review and editing; Roman Franek, Investigation, Writing – review and editing; Itamar Harel, Conceptualization, Resources, Data curation, Formal analysis, Supervision, Funding acquisition, Validation, Investigation, Visualization, Methodology, Writing – original draft, Project administration, Writing – review and editing

## Author ORCIDs

Eitan Moses http://orcid.org/0000-0003-0090-335X
Roman Franek http://orcid.org/0000-0002-3464-1872
Itamar Harel https://orcid.org/0000-0001-9749-8279

## Ethics

All turquoise killifish care and uses were approved by the Subcommittee on Research Animal Care at the Hebrew University of Jerusalem (IACUC protocols #NS-18-15397-2 and #NS-22-16915-3).

## Decision letter and Author response

Decision letter https://doi.org/10.7554/eLife.85960.sa1
Author response https://doi.org/10.7554/eLife.85960.sa2

# Additional files

## Supplementary files

• MDAR checklist

## Data availability

All plasmids and corresponding annotated maps are available via Addgene. All fish lines are available upon request. All raw images and datapoints used to generate the presented graphs have been submitted as source data.

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
