## [Editor Report]

Moses and Harel generate a compelling set of novel molecular tools in African turquoise killifish, including the development of gain of function and dose dependent inducible expression systems for African turquoise killifish. These tools will help boost this budding model system for broad biotechnological applications, including the study of gene function in the context of aging in a relatively fast manner compared to other vertebrate models. The authors showcase the efficacy of their tools in the context of peptide hormones involved in growth and gonad development.

---

## [Decision Letter]

**Decision letter after peer review:**

Thank you for submitting your article "A scalable platform for functional interrogation of peptide hormones in fish" for consideration by *eLife*. Your article has been reviewed by 3 peer reviewers, including Dario Riccardo Valenzano as the Reviewing Editor and Reviewer #1, and the evaluation has been overseen by Didier Stainier as the Senior Editor. The following individual involved in review of your submission has agreed to reveal their identity: Eve Seuntjens (Reviewer #3).

Essential revisions:

1) The authors should validate their rescue model better (timing, expression levels, duration of rescue, whole body or local), including proper controls.

2) The authors should provide a deeper characterization for the electroporation and Dox inducible system (see reviewer #3).

3) The authors need to thoroughly revise the English in the written text.

4) The manuscript does not obviously follow a question-driven flow and the authors do not make a compelling case about the necessity of developing such platform.

The manuscript could be framed as a tool/resource, showcasing the interventions with gh and fshb to support the tool.

*Reviewer #1 (Recommendations for the authors):*

While I am a big fan of this work, I find the manuscript not thoroughly edited. There are several sentences that are hard to understand, starting with the abstract.

Authors should check and review extensively for improvements to the use of English. The abstract contains several arbitrary (non substantiated) statements ("…particularly in females"' "As killifish 20 maturation is germline-independent,…").

Overall, I find this manuscript more as a container of successfully developed tools (about which I congratulate the authors) for the killifish community.

I wonder whether *eLife* is the right fit for this work.

I find a large disconnect between the way in which the manuscript is written and the quality of the figures. While the figures have a very high quality standard, Abstract, Introduction and Discussion are not doing justice to the work done, as they appear to be in an early draft format.

Scientific questions to address:

– Do gh1-d4/d4 display embryos of similar size to the wt (line 77-80)?

– "our findings demonstrate that GH deficiency delays both somatic growth and maturity". Was this known? Is this new? It's not clear what was the initial hypothesis. There seems to be no clear hypothesis to justify the experiment. The hypothesis needs to be spelled out more clearly.

– Is the approach generally new? Or, rather, is the approach new in the specified model system (killifish)?

– In figure 4: what is labelled as "M"? I see that "M" also appears in Figure 2. However, "mature" is an adjective, not a noun. The authors need to explain more clearly what is this structure.

– line 176-177: "Our method is compatible with multiplexing". Do the authors know this already, have they multiplexed interventions or is this a speculation?

*Reviewer #2 (Recommendations for the authors):*

It is clear from Figure 2B that the gh1 deletion has a drastic impact on fertility in addition to somatic growth. The number of embryos per collection appears to be significantly lower as well. It is only mentioned in passing in the manuscript, but this aspect should be characterized and discussed in depth.

The statistics in Figure 2B are unclear. Are P-values from weekly pairwise comparison? If this is the case, the stages being compared are different because the mutant matures later, so I'm not sure what the statistics mean. If the goal is to compare delay in the onset of peak fertility, the statistics should be between the time it takes to reach the peak fertility. Alternatively, a different comparison based on the objective should be done.

The sections in Figure 2C appear to be from different parts of the mutant and WT ovaries. I am not an expert in histology, but I find it difficult to see the similarities between the two conditions. Can the authors provide a zoomed-out image and ensure that all the images used throughout the manuscript come from the same regions?

It is unclear if the authors detect endogenous or injected gh because electroporation is performed on WT fish (lines 97-103), which should have gh already. Therefore, WT injection is not a good controls for the electroporation efficiency. First, it needs to be established that the mutant has no gh. An appropriate control for electroporation will be injecting gh plasmid construct in gh1-/- fish, and quantifying gh protein in isolated muscle before and after electroporation.

A control GFP image after injection but before electroporation is also needed in Figure 3A.

The rescue experiment also needs a bit more characterization. First, Figure 3B needs the size data from mutant fish right before electroporation and after the increase in size. Second, pictures of the fish before and after the size increase should be included. Third, gh has a growth-promoting effect systematically on all tissues, including bones, cartilage, etc., which raises several interesting questions. Is the gh from the injected construct also transported throughout the body? Can it be detected in other tissues and organs? If not, what is the mechanism of the growth increase? It is important to characterize or discuss how the size increase is accomplished. Finally, it should be assessed if there is rescue of embryo production in injected fish (Figure 2B). Or is the effect restricted to growth only?

The rescue of the infertility phenotype due to fshb loss of function is interesting. The authors say that the plasmid is stably expressed for several weeks. Is the expression localized to muscle, or is fshb transported and detectable in the ovaries, where it is expected to act (Figure 4)?

The dose dependent gh1 expression needs to be quantified in dissected muscle using the gh1 antibody. GFP image is not convincing as a proxy for gh1 expression without proper controls (see above). Also, does this rescue the gh loss of function phenotype in a dose dependent manner?

The Dox inducible system is the most interesting part of this work. But in its current form, it is the most under-developed. It is unclear which gene is rescued here. And does it also rescue the associated phenotype in a dose dependent manner?

My final point concerns the study's diffused focus. The authors state that they are working on another manuscript with the detailed characterization of the defects due to fshb and other pituitary hormones, which is fine. But likely due to that, the current manuscript lacks details in the characterization of both the methods and the biology of phenotypes. In my opinion, it will be better to focus on one or two genes but thoroughly characterize every aspect of the tools and biology.

*Reviewer #3 (Recommendations for the authors):*

Validation of the knockout models

For all hormones tested, are the genes you chose the only copies in the genome? Is there only 1 gh, fshb, tshb gene or are there duplicates? Have you measured plasmid concentrations of these hormones to validate the fact they are mutant? Or alternative ways to validate absence of hormone at protein level?

Rescue modalities

Rescue was induced by a single electroporation using a CMV-driven plasmid. How did you decide on the age of rescue? Have you tested different ages of rescue induction? If not, can you speculate on the outcome of such an experiment? Did you include a control where an empty plasmid was electroporated in the mutants? This would be the correct additional control to calculate your rescue percentage. The latter would be important for gh, where you seem to have a sex-specific difference in rescue efficiency. The females do not seem to be rescued efficiently: please adapt lines 109-110: this conclusion might only hold for male fish.

How long does overexpression last? Is this a lifelong effect?

Regarding the "dose-response" in Figure 5B: Can you quantify this? Is there a limit to the overexpression? Electroporations are notoriously variable: what was the variability of your system? Can you comment?

Doxycycline system

Here, you use EF1a as a general promoter: how efficient is this promoter compared to CMV? The GFP signal you observe as a readout seems to be rather limited in expression compared to the previous overexpression systems: can you benchmark this system better? For instance can you rescue the gh or fshb ko phenotype using the doxycycline construct to prove the system is comparably efficient?

---

## [Author Response]

Essential revisions:Reviewer #1 (Recommendations for the authors):While I am a big fan of this work, I find the manuscript not thoroughly edited. There are several sentences that are hard to understand, starting with the abstract.Authors should check and review extensively for improvements to the use of English. The abstract contains several arbitrary (non substantiated) statements ("…particularly in females"' "As killifish 20 maturation is germline-independent,…").Overall, I find this manuscript more as a container of successfully developed tools (about which I congratulate the authors) for the killifish community.I wonder whether eLife is the right fit for this work.I find a large disconnect between the way in which the manuscript is written and the quality of the figures. While the figures have a very high quality standard, Abstract, Introduction and Discussion are not doing justice to the work done, as they appear to be in an early draft format.Scientific questions to address:– Do gh1-d4/d4 display embryos of similar size to the wt (line 77-80)?

Yes, we have now measured the size of the eggs from both *gh1* homozygous and heterozygous parents, and found that they are comparable to WT fish (see new Figure 2—figure supplement 1B).

– "our findings demonstrate that GH deficiency delays both somatic growth and maturity". Was this known? Is this new? It's not clear what was the initial hypothesis. There seems to be no clear hypothesis to justify the experiment. The hypothesis needs to be spelled out more clearly.

We agree with the reviewer that this was not clear. As the manuscript was originally submitted as a ‘Tools and Resources’ article, the narrative used the *gh1* and *fshb* mutations, primarily to support the gain-of-function tool that we have developed. to support the tool. However, we now describe that several phenotypes of hormonal perturbations in killifish are more similar to mouse, and not to zebrafish and medaka. We believe this might be related to the germline independent sexual differentiation. We now discuss this throughout the text, including:

“Based on this mammalian-like trait, we hypothesize that hormonal perturbations in killifish might produce phenotypes that more faithfully recapitulate the corresponding mouse models, than do similar mutations observed in zebrafish.”

We aimed to develop a reversible control of hormone function, focusing on evolutionary conserved pathways. Thus, allowing us to better generalise our findings to other vertebrate species. We now indicate the following in the introduction:

“The subsequent phenotypes indicate that although the killifish undergoes rapid growth and puberty, it still follows a vertebrate-conserved genetic program. This allows us to use our data to improve the understanding of how these hormones regulate the onset and duration of specific traits across evolutionary distances.”

– Is the approach generally new? Or, rather, is the approach new in the specified model system (killifish)?

To the best of our knowledge, this is the first time an intramuscular electroporation system is used to functionally rescue hormone deficiencies in a genetic model (practically using the muscle as a body-wide “factory”).

– In figure 4: what is labelled as "M"? I see that "M" also appears in Figure 2. However, "mature" is an adjective, not a noun. The authors need to explain more clearly what is this structure.

We thank the reviewer for bringing this to our attention. For simplicity, we followed common nomenclature in the field, which broadly uses “M” to indicate this stage (see: “Lubzens et al., Oogenesis in teleosts: How fish eggs are formed, General and Comparative Endocrinology, 2010).

Nonetheless, we have now clearly marked all the developmental stages of oocytes and sperm, in the figures and figure legends (specifically in Figures 2 and 4).

– line 176-177: "Our method is compatible with multiplexing". Do the authors know this already, have they multiplexed interventions or is this a speculation?

We have now generated a new vector (*fshb-T2A-dTomato*)*,* and demonstrated the feasibility for multiplexing, showing that when 2 vectors tagged with different fluorophores are injected into the fish, individual colors can be detected.

Reviewer #2 (Recommendations for the authors):It is clear from Figure 2B that the gh1 deletion has a drastic impact on fertility in addition to somatic growth. The number of embryos per collection appears to be significantly lower as well. It is only mentioned in passing in the manuscript, but this aspect should be characterized and discussed in depth.

We now extensively characterize the *gh1* phenotypes. Specifically, we have imaged the dissected gonads of WT fish, *gh1* mutants, and rescued *gh1* mutants (see new Figure 3F). We have also imaged transverse histological sections showing the entire body cavity of WT and gh mutant females (see new Figure 2—figure supplement 1A). Using these images, we suggest that a constraint on fertility could be simple space. Specifically, while egg size does not change (see new Figure 2—figure supplement 1B), and the fish size does change, space for mature eggs is restricted. Additionally, we have characterised the effect of the *gh1* rescue on reproduction (see new Figure 3E) and size of the ovary (see new figure 3F) and show there is both a rescue of fertility and an accompanying increase in the ovary size. We discuss all of this extensively throughout the text.

The statistics in Figure 2B are unclear. Are P-values from weekly pairwise comparison? If this is the case, the stages being compared are different because the mutant matures later, so I'm not sure what the statistics mean. If the goal is to compare delay in the onset of peak fertility, the statistics should be between the time it takes to reach the peak fertility. Alternatively, a different comparison based on the objective should be done.

We agree with the reviewer this might be confusing. Therefore, we have now added two new graphs to more directly compare specific parameters: (a) The difference in the timing of peak fertility (temporal); and (b) the difference of fecundity at peak fertility (reproductive output) (new Figure 2C). In addition, we think that presenting the comparison of the WT and mutant fish at the same biological age (Figure 2B) might be still visually informative, and retains biological meaning.

The sections in Figure 2C appear to be from different parts of the mutant and WT ovaries. I am not an expert in histology, but I find it difficult to see the similarities between the two conditions. Can the authors provide a zoomed-out image and ensure that all the images used throughout the manuscript come from the same regions?

The reviewer is correct. While both pictures did show oocytes from comparable regions, they were at different stages of development. We now display in Figure 2D oocytes at similar stages, and have labelled these stages accordingly. We also provided a zoomed-out image (new Figure 2—figure supplement 1A).

It is unclear if the authors detect endogenous or injected gh because electroporation is performed on WT fish (lines 97-103), which should have gh already. Therefore, WT injection is not a good controls for the electroporation efficiency. First, it needs to be established that the mutant has no gh. An appropriate control for electroporation will be injecting gh plasmid construct in gh1-/- fish, and quantifying gh protein in isolated muscle before and after electroporation.

We now electroporate *gh1* homozygous mutants with either a *CMV:GFP* plasmid, a *CMV:gh1-T2A-GFP* plasmid (at 2 different concentrations, see comment #9), or no plasmid at all, and perform immunohistochemistry on these tissue sections (new Figure 3—figure supplement 1A).

A control GFP image after injection but before electroporation is also needed in Figure 3A.

We now provide images for fish injected with *CMV:GFP* and imaged 3 days after injection, with or without electroporation (See new Figure 3B).

The rescue experiment also needs a bit more characterization. First, Figure 3B needs the size data from mutant fish right before electroporation and after the increase in size. Second, pictures of the fish before and after the size increase should be included.

We have now measured the rescued *gh1* mutant fish right before electroporation and after the increase in size. The results are provided for both sexes (all comparisons together are available in Author response image 1).

**Author response image 1. sa2fig1:** (A) Quantification of somatic growth (standard length) of 12-week-old WT fish compared to *gh1^Δ4/+^* mutants, *gh1^Δ4/Δ4^* mutants following electroporation of a *gh1-T2A-GFP* plasmid, controls injected with an empty vector and 8-week-old fish before injection. males (left) and females (right), n ≥ **3** individuals from each experimental group. Error bars show mean ± SEM. Significance was calculated using oneway ANOVA with a Dunnet post-hoc compared to the WT. Exact p-values are indicated. (B) A phylogenetic tree based on the GH, LHB (another pituitary hormone in the reproductive axis), TSHB, and FSHB proteins of Humans (*Homo Sapienes)*, Mice (*Mus musculus)*, Chickens (*Gallus gallus)*, Japanese Medaka (*Oryzias latipes*), Nile Tilapia (*Oreochromis niloticus*), Zebrafish (*Danio rerio*), Salmon (*Salmo salar*), and the African Turquoise Killifish (*Nothobranchius furzeri*).

Mutant fish slightly grow in size, even in the absence of GH intervention. Therefore, we believe that a more intuitive control is through a comparison with age-matched interventions. Thus, leaving only one parameter to compare (the effect of the rescue, independent on natural growth). Accordingly, we now use fish that were electroporated with a *CMV:GFP* plasmid, as our new control, and provide their representative images and size measurements (compared to fish that were electroporated with the *gh1* plasmid, new Figure 3D).

Third, gh has a growth-promoting effect systematically on all tissues, including bones, cartilage, etc., which raises several interesting questions. Is the gh from the injected construct also transported throughout the body? Can it be detected in other tissues and organs? If not, what is the mechanism of the growth increase? It is important to characterize or discuss how the size increase is accomplished. Finally, it should be assessed if there is rescue of embryo production in injected fish (Figure 2B). Or is the effect restricted to growth only?

First, as the GH antibody did not work well in WB, we chose several alternatives. As this is also relevant to the next comment, concerning *fshb*, we selected *fshb* as a proof-of-principle to measure the body-wide distribution of an electroporated hormone.

Specifically, we generated a new construct of FSHB-GFP fusion protein, and demonstrate that this construct was able to fully rescue *fshb* mutant infertility (see new Figure 4—figure supplement 1B). While some GFP signal was observed in the gonads following muscle electroporation, it was not robust (see new Figure 4figure supplement 1C). This could be due to the extremely low amount of hormone required for a functional effect, and is now extensively characterised using the Dox system (new Figure 5).

In regard to the body-wide effect, we now demonstrate that the growth of GH-rescued fish maintains the correct body proportions (length vs. width) (See new Figure 3—figure supplement 1B), as well as completely rescues fertility output (see new Figure 3E). This is in contrast to altered body proportions observed, for example, in the *tshb* mutants (see new Figure 4C). These data support that notion that the rescue affects multiple systems and organs. Similarly, GH therapy in humans is administered systemically by injection, and effects are body-wide through the circulation. This is now discussed in the text, for example:

“Fish retain normal body dimensions (Figure 3—figure supplement 1B), suggesting uniform body-wide hormone effect.”

The rescue of the infertility phenotype due to fshb loss of function is interesting. The authors say that the plasmid is stably expressed for several weeks. Is the expression localized to muscle, or is fshb transported and detectable in the ovaries, where it is expected to act (Figure 4)?

We generated a new construct of FSHB-GFP fusion protein, and demonstrate that while some GFP signal was observed in peripheral organs (i.e. the gonads) following muscle electroporation, it was not robust (see new Figure 4—figure supplement 1C). This could be due to the extremely low amount of hormone required for a functional effect, and is now functionally characterised using the Dox system (new Figure 5).

The dose dependent gh1 expression needs to be quantified in dissected muscle using the gh1 antibody. GFP image is not convincing as a proxy for gh1 expression without proper controls (see above). Also, does this rescue the gh loss of function phenotype in a dose dependent manner?

We now demonstrate the staining of GH in *gh1* mutant fish (with or without electroporation, at 2 different doses, new Figure 3A).

Due to the extremely small size of GH fish, experiments using large numbers of mutants, are more technically challenging. Therefore, alternatively, to test the dose-dependent physiological effect of our platform, we used the FSHB knock-outs. Specifically, we mated rescued female *fshb* homozygous fish, either injected with 5μg or 0.5μg of the *fshb-GFP* plasmid, and measured fertility. The dose did not affect the fertility output (see new Figure 5C). We think this is because very low amounts of the hormone are sufficient for complete rescue. This is also supported by the fact that heterozygous fish demonstrate comparable fertility to WTs (see new Figure 4—figure supplement 1A). To further explore this, we generated an inducible construct (see next comment), which provided better control of hormone levels, by switching production on and off, and assessing fertility at extremely low hormone concentrations (new Figure 5).

The Dox inducible system is the most interesting part of this work. But in its current form, it is the most under-developed. It is unclear which gene is rescued here. And does it also rescue the associated phenotype in a dose dependent manner?

We have now constructed a Dox-dependent *fshb* system, and excitingly, demonstrate that fertility could be reversibly switched on and off. Additionally, it appears that minor leakiness of the Dox system is sufficient to restore limited fertility. This further highlights the low concentration of FSHB required for a functional effect (e.g. the functional threshold is lower than the detected GFP signals) (see new Figure 5E, F).

My final point concerns the study's diffused focus. The authors state that they are working on another manuscript with the detailed characterization of the defects due to fshb and other pituitary hormones, which is fine. But likely due to that, the current manuscript lacks details in the characterization of both the methods and the biology of phenotypes. In my opinion, it will be better to focus on one or two genes but thoroughly characterize every aspect of the tools and biology.

As the manuscript was originally submitted as a ‘Tools and Resources’ article, the narrative highlighted primarily the gain-of-function interventions. We agree with the reviewer and now expand the characterization of the mutant and rescue phenotypes, as well as the ability to finely control expression using our tools. Accordingly, the narrative of the entire manuscript is now altered.

Reviewer #3 (Recommendations for the authors):Validation of the knockout modelsFor all hormones tested, are the genes you chose the only copies in the genome? Is there only 1 gh, fshb, tshb gene or are there duplicates? Have you measured plasmid concentrations of these hormones to validate the fact they are mutant? Or alternative ways to validate absence of hormone at protein level?

To address this, we have built a phylogenetic tree showing that all selected genes are more closely related to their orthologs than to other genes in the family. Thus, confirming the annotation (Author response image 1). In addition, the evolutionary-conserved phenotypes of our genetic models imply there is no redundancy. Because of the low quality of antibodies in killifish (and fish in general) we were unable to validate the mutants on the protein level (when not overexpressed via electroporation, as seen for GH). We hope that the functional assays and phenotypic rescue mitigate this challenge.

Rescue modalitiesRescue was induced by a single electroporation using a CMV-driven plasmid. How did you decide on the age of rescue? Have you tested different ages of rescue induction? If not, can you speculate on the outcome of such an experiment? Did you include a control where an empty plasmid was electroporated in the mutants? This would be the correct additional control to calculate your rescue percentage. The latter would be important for gh, where you seem to have a sex-specific difference in rescue efficiency. The females do not seem to be rescued efficiently: please adapt lines 109-110: this conclusion might only hold for male fish.

While the younger the better, the specific age of intervention in *gh1* mutants was selected since at a younger age these fish (which are extremely small) are more fragile, and therefore died from the procedure itself. We can speculate that the remaining time to grow could have an effect. We now discuss this in the text:

“The partial rescue in somatic growth might be due to the age at intervention. Specifically, electroporation was performed at 8 weeks of age, which is already past the period during which the fish exhibit a growth spurt and the onset of puberty. This timepoint was selected because of the difficulties in injection and electroporation in the extremely small young mutants. In contrast to the partial growth rescue, reproductive defects were fully rescued in GH-electroporated mutant couples (Figure 3E), as evidenced by an increase in the size of the ovary (Figure 3F). Together, these results demonstrate that ectopic muscle electroporation can rescue many aspects of GH deficiency. “

Additionally, we have now added the empty plasmid injected controls. With these controls, we now demonstrate that while both sexes are similarly affected (i.e. increase in average length), males (which grow to be much larger), probably provide more dynamic range to detect statistically significant growth changes. See new Figure 3D.

How long does overexpression last? Is this a lifelong effect?

We have now functionally tested the fertility of *fshb* mutant females injected with 5μg or 500ng of an fshbgfp plasmid at the age of 1 month. We show that the fertility of both concentrations is comparable to WT females for at least 2 months (see new Figure 5D). As at the age of 4 months both WT and rescued fish experience age-related decline in fertility, we terminated the experiment. Importantly, we now further demonstrate precise temporal control by using the Dox system, in which a switch-on and off can be achieved in a matter of 2-3 weeks.

Regarding the "dose-response" in Figure 5B: Can you quantify this? Is there a limit to the overexpression? Electroporations are notoriously variable: what was the variability of your system? Can you comment?

To quantify electroporation variability, we have analysed the GFP intensity in images from two doses of plasmid. Though there is some variability within each concentration, the difference between concentrations is significant (see new Figure 5D).

The overexpression is limited by plasmid concentration. While beyond the scope of this manuscript, we speculate that multiple injections are theoretically possible, though our data indicated that saturation is already reached al lower concentrations (see new Figure 5E).

Doxycycline systemHere, you use EF1a as a general promoter: how efficient is this promoter compared to CMV? The GFP signal you observe as a readout seems to be rather limited in expression compared to the previous overexpression systems: can you benchmark this system better? For instance can you rescue the gh or fshb ko phenotype using the doxycycline construct to prove the system is comparably efficient?

Indeed, CMV is stronger than Ef1a. Accordingly, we now show that we are able to obtain comparable GFP signals using the *Ef1a:Dox* system, simply by increasing the plasmid concentration (see new panel in Figure 5E). Additionally, we now perform an *fshb* rescue under dox control, see new Figure 5F.